# [Re] Boosting the Visual Interpretability of CLIP via Adversarial Fine-Tuning

## Abstract

This paper presents a reproducibility study of "Boosting the Visual Interpretability of CLIP via Adversarial Fine-Tuning" by Gong et al. (2025), published at ICLR 2025, which proposes an unsupervised adversarial fine-tuning (AFT) method with norm regularization to enhance the visual interpretability of CLIP's image encoder. We attempt to reproduce the key claims regarding improved saliency map quality, increased concept alignment, transferability to out-of-distribution datasets, and the trade-off with zero-shot accuracy. Beyond reproduction, we propose a saliency-guided regularization extension that introduces an Energy Pointing Game loss, directly supervising the spatial alignment of Simple Gradient saliency maps with target objects. We evaluate our extension across a range of saliency-loss weights and show that explicit saliency supervision improves localization metrics with only a modest reduction in adversarial robustness.

## 1 Introduction

CLIP (Contrastive Language-Image Pre-training) (Radford et al., 2021) has become a foundational model for vision-language representation learning and is widely used as a backbone in modern multimodal systems. Despite its strong performance, the internal representations of CLIP's image encoder remain difficult to interpret. In particular, common gradient-based attribution methods (e.g., Simple Gradient and Grad-CAM (Selvaraju et al., 2017)) often produce noisy and unstable saliency maps, and the encoder itself appears to contain only a limited number of human-interpretable concept detectors.

To understand why these saliency maps are unreliable, it helps to recall how they are constructed. Gradient-based attribution methods quantify the sensitivity of model's output to small input perturbations. For a model $f_\theta$ with parameters $\theta$, the Simple Gradient saliency map (Simonyan et al., 2013) is defined as $S(x) = |\nabla_x \mathcal{L}(f_\theta(x), y)|$, where $\mathcal{L}$ is a loss function. However, these gradients in pretrained CLIP models tend to be dense and noisy, distributing attribution broadly across the input rather than focusing on semantically meaningful regions.

Gong et al. (2025) address the problem of noisy, diffuse gradients in pretrained CLIP by proposing adversarial fine-tuning (AFT) with norm regularization, which enforces sparse encoding of visual features. They provide a theoretical analysis showing that AFT implicitly regularizes input gradients, leading to more concise and interpretable saliency maps, and their experiments demonstrate significant improvements across multiple interpretability metrics. However, because AFT relies solely on the adversarial objective, it does not explicitly enforce alignment between saliency maps and semantically meaningful regions of the input.

In this work, we conduct a thorough reproducibility study of Gong et al. (2025), verifying their central claims about saliency map quality, concept alignment, transferability, and the accuracy-robustness trade-off, and extend their methodology with explicit spatial supervision. Our main contributions are:

- **Reproduction of the main claims.** We independently verify the four central claims of the original paper regarding saliency map quality, neuron-concept alignment, transferability to out-of-distribution datasets and downstream tasks, and the trade-off between zero-shot accuracy and adversarial robustness.

- **Implementation of evaluation protocols.** Since the original repository does not release code for a broad range of evaluation strategies, we implement these evaluations from scratch and release them as part of our reproducible codebase.

- **Saliency-guided regularization extension.** We propose a novel training objective that augments AFT with an Energy Pointing Game loss computed over COCO instance masks, directly supervising the spatial alignment of Simple Gradient saliency maps with foreground objects. We show that explicit saliency supervision improves localization metrics while preserving most of the adversarial robustness provided by AFT.

## 2 Scope of Reproducibility

All our experiments aim to demonstrate the improvement in model interpretability achieved by training with the proposed AFT method. To this end, we focus on reproducing the following main claims from Gong et al. (2025):

- **Claim 1 (Saliency Map Quality):** AFT significantly improves the quality of Simple Gradient and Grad-CAM saliency maps, as measured by Pointing Game accuracy and related localization metrics (Table 1, Figure 2a,b in the original paper).

- **Claim 2 (Concept Alignment):** AFT increases neuron-concept alignment in CLIP's visual encoder.

- **Claim 3 (Transferability):** The interpretability improvements transfer to out-of-distribution datasets and downstream tasks, such as linear probing (Figure 2c,d, Figure 3 in the original paper).

- **Claim 4 (Accuracy Trade-off):** AFT maintains reasonable zero-shot classification accuracy while improving interpretability, with only a 2.84% drop at $\epsilon = 1/255$ (Table 2 in the original paper).

Each experiment in Section 4 is designed to evaluate at least one of these claims.

Beyond these reproduction targets, we propose and evaluate an extension: a saliency-guided regularization loss that explicitly aligns Simple Gradient attribution maps with ground-truth segmentation masks during adversarial fine-tuning. While the original paper demonstrates that AFT implicitly regularizes input gradients through its norm-regularized minimax formulation, it does not explicitly enforce spatial alignment between saliency maps and semantically meaningful regions of the input. We therefore investigate whether adding explicit spatial supervision via ground-truth masks can further improve saliency map quality without sacrificing adversarial robustness.

## 3 Methodology

To reproduce the study, we use the publicly available repository provided by the authors. The original codebase included training scripts for multiple model backbones (ResNet-50, ViT-B/16, and ViT-L/14), implementations of adversarial attacks (PGD (Madry et al., 2018) and APGD (Croce & Hein, 2020b)), as well as utilities for saliency map visualization (Simple Gradients and Grad-CAM).

We restructure the codebase to improve modularity and configurability, enabling more systematic and flexible experimentation. We also extend the implementation to include the full set of evaluation experiments supporting the main claims of the original study. The original repository does not include code for dataset-level evaluation, evaluation under adversarial attacks, or for computing key metrics such as Pointing Game (PG), energy-based PG, Pixel Accuracy (PA), Average Precision (AP), and Intersection over Union (IoU). We therefore implement these components from scratch. In addition, we implement the neuron-concept alignment evaluation based on Oikarinen & Weng (2023).

Moreover, we introduced complementary evaluations, including retrieval performance measured by Recall@k for both text-to-image ($T \rightarrow I$) and image-to-text ($I \rightarrow T$) tasks.

Finally, we verify that the identified salient regions are causally relevant through ROAR (Remove And Retrain) analysis (Hooker et al., 2019), which measures accuracy degradation when removing the most salient pixels.

Our reproduction code is available at `https://anonymous.4open.science/r/reboosting-CLIP`

## 3.1 Unsupervised AFT and Optimization

The original paper first formulates supervised AFT as a minimax objective over text–image similarity (Equation 3 in Appendix B). Since the text encoder is frozen, the authors derive an image-only upper bound, yielding the unsupervised AFT objective:

$$\min_{\theta} \mathbb{E}_{x \sim \mathcal{D}_{\text{train}}} \max_{\delta_x} \left[ \frac{1}{2} \left\| \mathbb{E}_{z \sim \mathcal{N}(0, \sigma^2 I)}[f_\theta(x + z + \delta_x)] - I_x \right\|^2 - h(\delta_x) \right], \tag{1}$$

where $f_\theta$ is the trainable visual encoder, $x$ is a clean training image, and $I_x$ is the reference embedding produced by the frozen original CLIP image encoder. The perturbation $\delta_x$ is constrained within an $\ell_\infty$ budget $\varepsilon$, $z \sim \mathcal{N}(0, \sigma^2 I)$ is Gaussian smoothing noise, and $h(\delta_x)$ penalises large perturbations. Thus, the inner maximisation finds a worst-case perturbation that moves the perturbed embedding away from $I_x$, while the outer minimisation trains $f_\theta$ to preserve the reference embedding under this perturbation. We adopt this image-only objective because it is the primary formulation used in the original experiments.

The inner maximisation in Equation 1 is approximated with Projected Gradient Descent (PGD) (Madry et al., 2018). Starting from $\delta_x^{(0)} = 0$, the perturbation is updated for $T_{\text{adv}}$ steps as

$$\delta_x^{(t+1)} = \Pi_\varepsilon \left( \delta_x^{(t)} + \alpha \nabla_{\delta_x} \mathcal{L}(\delta_x^{(t)}) \right), \tag{2}$$

where $\mathcal{L}$ is the inner objective in Equation 1, $\alpha$ is the PGD step size, and $\Pi_\varepsilon$ projects the perturbation back onto the $\ell_\infty$ ball of radius $\varepsilon$. The final perturbation $\delta_x^{(T_{\text{adv}})}$ is then used for the outer update of $\theta$.

## 3.2 Hyperparameters

**Model description.** Following Appendix A.3 of the original paper, we use ViT-B/16 (Dosovitskiy et al., 2021) for all experiments, except for the evaluation with ROAR, which is conducted using the ResNet-50 (He et al., 2015) architecture. For the ViT-B/16 baseline, we use OpenCLIP (Cherti et al., 2023). During AFT, only the visual encoder is fine-tuned while the text encoder remains frozen. Due to resource constraints, we focus our reproduction on ViT-B/16, which the original paper uses for the majority of its experiments (ViT-L/14 is used primarily for zero-shot accuracy).

**Adversarial Fine-Tuning Setup.** Following Appendix A.3 of the original paper, we train the ViT-B/16 model with PGD on the ImageNet2012 dataset (Deng et al., 2009). We train two separate models with different perturbation strengths, $\epsilon \in \{1/255, 4/255\}$. The adversarial attack is performed in the $\ell_\infty$ norm with 10 PGD steps and a step size of $1/255$.

Training is carried out for 20,000 iterations with a linear warmup of 1,400 steps followed by a cosine learning rate decay. We use the AdamW optimizer with $\beta_1 = 0.9$ and $\beta_2 = 0.95$, a peak learning rate of $1 \times 10^{-5}$, and a weight decay of $1 \times 10^{-4}$. The training objective follows the unsupervised AFT formulation in Equation 1, where the embedding-consistency term is implemented with an $\ell_2$ loss; training is performed with a batch size of 128. Following the original paper, we set the Gaussian smoothing parameter $\sigma = 1/255$ and the Huber smoothness parameter $\eta = 1/255$. All experiments are conducted on two NVIDIA A100 GPUs(40GB each).

A complete summary of all datasets used in our experiments is provided in Appendix A.

### 3.3 Saliency-Guided Regularisation

We introduce an auxiliary loss term that directly supervises the spatial alignment of gradient-based saliency maps with reference object masks. The key idea is to measure the concentration of input-gradient magnitude within annotated foreground regions, penalising the model when saliency mass falls on background features.

**Saliency construction.** Following Gong et al. (2025), given a clean input image $x$, we compute the Simple Gradient saliency map as the input gradient of the *feature-consistency score*

$$s = I_{orig}^{\top} I_x,$$

the cosine similarity between the trainable embedding $I_x = f_{\theta}(x)$ and a frozen reference embedding $I_{orig} = f_{orig}(x)$ from OpenAI CLIP (both unit-normalised). We then compute the input gradient $g = \nabla_x s$ using `create_graph=True`, ensuring the gradient remains part of the differentiable computational graph. This enables the calculation of second-order derivatives during the backward pass of the saliency loss. The final saliency map $S$ is constructed by aggregating the absolute gradient magnitudes across the channel dimension:

$$S = \sum_{c=1}^{C} |g_c|,$$

where $C$ is the number of input channels ($C=3$ for RGB in our experiments) and $g_c$ is the $c$-th channel of $g$. The resulting heatmap has shape $(B, H, W)$.

**Energy Pointing Game loss (EPG-loss).** We first normalise the saliency map into a spatial probability distribution for each image,

$$\hat{S}_i(p) = \frac{S_i(p)}{\sum_{p'} S_i(p')},$$

where $p$ indexes pixel locations in the $H \times W$ spatial grid and $\sum_{p'}$ runs over all such locations. Let $M_i \subseteq \{1, \ldots, H\} \times \{1, \ldots, W\}$ be the binarised foreground mask of image $i$ from COCO instance annotations. The EPG loss is then

$$\mathcal{L}_{\text{EPG}} = 1 - \frac{1}{B} \sum_{i=1}^{B} \sum_{p \in M_i} \hat{S}_i(p).$$

By construction, $\mathcal{L}_{\text{EPG}} \in [0, 1]$: $\mathcal{L}_{\text{EPG}} = 0$ indicates that all saliency mass is concentrated inside the target mask, whereas $\mathcal{L}_{\text{EPG}} = 1$ indicates that no saliency mass falls within it.

**Integration into training.** The full training objective is

$$\mathcal{L}_{\text{total}} = \mathcal{L}_{\text{adv}} + w_s \mathcal{L}_{\text{EPG}},$$

where $w_s$ controls the contribution of the saliency regularization term. Since $\mathcal{L}_{\text{EPG}}$ is added independently of the clean/adversarial trade-off, its strength can be tuned separately, and setting $w_s = 0$ recovers the standard adversarial training objective. $\mathcal{L}_{\text{EPG}}$ also admits two complementary theoretical readings, as a mixed input/parameter Hessian regularizer, and as a background-restricted adversarial objective, both developed in Appendix E.

**Data.** We utilise the MSCOCO dataset (Lin et al., 2015), specifically leveraging its high-quality instance-level segmentation masks to provide ground-truth supervision for the saliency maps. Both images and their corresponding binary masks are resized to $224 \times 224$ pixels to match the input resolution of the CLIP vision transformer.

While the original AFT formulation of Gong et al. (2025) is entirely unsupervised, our saliency-guided extension introduces a supervised component by explicitly aligning the model's gradient-based attention with human-annotated foreground regions.

**Training configuration.** We found that training with the saliency loss from scratch does not converge. We attribute this to the second-order structure of the EPG gradient. Before AFT has converged, the input gradients $g_x$ are dense and near-uniform. The mixed input/parameter Hessian that drives $\nabla_{\theta} \mathcal{L}_{\text{EPG}}$ then has

random sign across pixels, producing high-variance updates that interfere with the adversarial objective. We formalize this argument in Appendix E. We therefore apply the saliency loss only as a fine-tuning stage on top of an already-converged checkpoint, where the adversarial objective has already established structured gradients that the EPG term can meaningfully reshape. Specifically, starting from an AFT[4] checkpoint trained for 5,000 steps on COCO, we fine-tune for 2,000 additional steps with a learning rate of $5 \times 10^{-6}$ and a warmup of 100 steps. To study the sensitivity of the method to the strength of the saliency regulariser, we sweep $w_s \in \{0, 0.2, 0.4, 0.6\}$. All configurations use a batch size of 128, which fits on 40 GB A100 GPUs.

## 4 Results

### 4.1 Results Reproducing Original Paper

#### 4.1.1 Saliency Map Quality (Claim 1)

**Qualitative Analysis.** We reproduced the qualitative saliency map comparison from the original paper (Gong et al., 2025) (Figure 2a,b), using our retrained ViT-B/16 model with $\epsilon = 4/255$. Following the original methodology, we selected images from the COCO validation set (Lin et al., 2015) containing multiple objects and generated saliency maps using different text prompts.

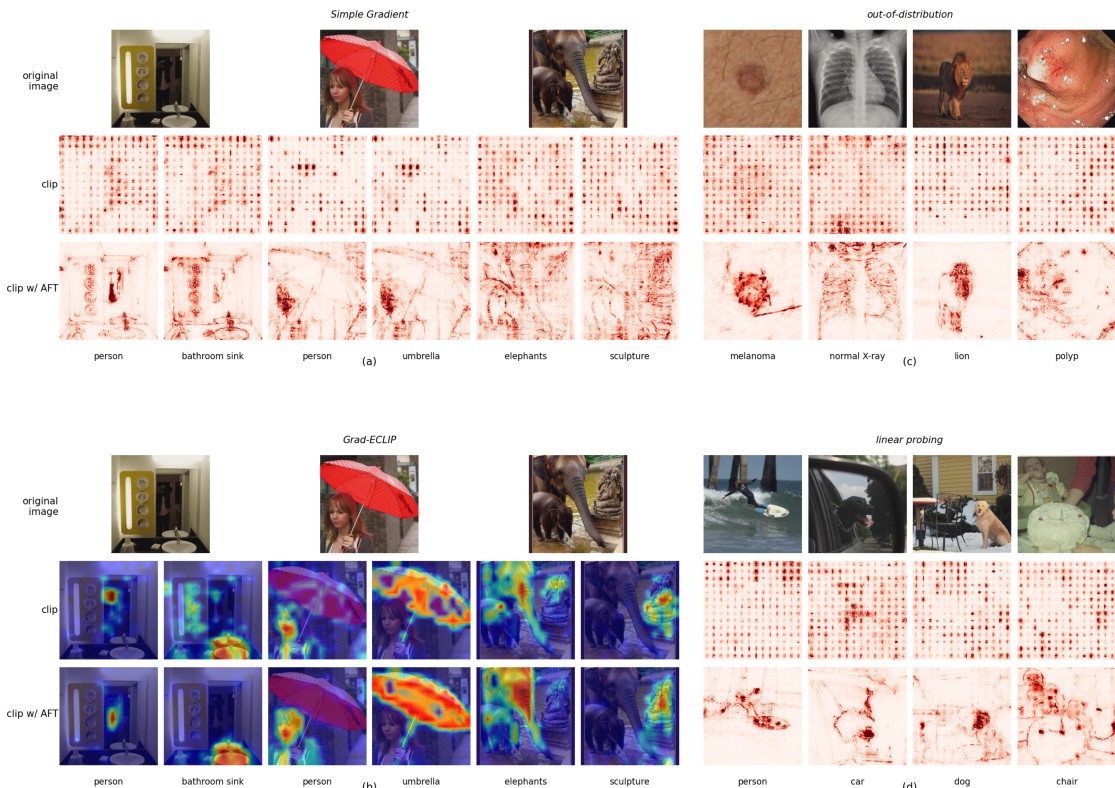

Figure 1: Saliency map comparison between CLIP (middle row) and CLIP with AFT (bottom row). (a) Simple Gradient maps for different text prompts. (b) Grad-ECLIP visualisations for the same images. (c) Out-of-distribution evaluation on medical and wildlife images. (d) Linear probing results on COCO classification.

Figure 1(a) shows Simple Gradient saliency maps for three image-prompt pairs. The original CLIP model produced dense, noisy gradients distributed broadly across the input, while CLIP with AFT generated substantially cleaner maps that concentrated on semantically relevant regions. We additionally evaluated Grad-ECLIP (Zhao et al., 2024), a CLIP-tailored variant of class activation mapping, shown in Figure 1(b). Consistent with the Simple Gradient results, AFT produced sharper, prompt-sensitive maps that localised

on the referenced object, while the baseline CLIP maps were diffuse and less prompt-sensitive. These visualisations closely resemble the qualitative results reported in the original paper and show the benefit of AFT. We also ran standard Grad-CAM for completeness, but found the results inconsistent. The observed inconsistency in Grad-CAM results likely stems from implementation limitations rather than model performance. As noted in the official repository, the Grad-CAM library generates unreliable outputs when applied to CLIP models, which are not natively designed for class activation mapping. Therefore, we consider the Grad-CAM evaluation to be inconclusive for reliably assessing the benefits of AFT.

Both Simple Gradient and Grad-ECLIP independently yield focused, prompt-sensitive saliency maps under AFT, closely matching the qualitative behavior reported in the original paper. Grad-CAM, by contrast, proved unreliable for CLIP-style models and produced mixed results, so we treat it as inconclusive rather than as counter-evidence. Overall, the qualitative analysis supports Claim 1.

**Quantitative Analysis.** We assess performance quantitatively using five localisation metrics, all computed against ground-truth segmentation masks: Pointing Game (PG), which measures whether the argmax saliency pixel falls inside the object mask; PG-Energy (Wang et al., 2020), the fraction of total saliency mass contained within the mask; Average Precision (AP), the area under the precision-recall curve treating saliency values as soft predictions; Pixel Accuracy (PA) and Intersection-over-Union (IoU), computed on binarised saliency maps. Several protocol details including the binarisation threshold, the strategy for upsampling patch-grid saliency maps to image resolution, and the hit-radius tolerance used in PG evaluation are not formally documented in the original work, and our choices for each are described below.

Table 1: Evaluation of localisation ability using the PG and PG-energy (Zhang et al., 2018) and segmentation metrics (PA, AP and Mask IoU) on the ImageNet-Segmentation validation dataset.

| Saliency | CLIP | Zero-shot classification | | | | | Linear probing | | | | |
|---|---|---|---|---|---|---|---|---|---|---|---|
| | | PG↑ | PG-energy↑ | AP↑ | Pix.Acc.↑ | IoU↑ | PG↑ | PG-energy↑ | AP↑ | Pix.Acc.↑ | IoU↑ |
| SG | w/o AFT | 9.53 | 34.19 | 35.91 | 58.15 | 13.29 | 25.79 | 29.64 | 31.74 | 61.28 | 12.32 |
| | w/ AFT[4] (ours) | **33.34** | **55.92** | **61.22** | 68.27 | **29.15** | 50.34 | 41.40 | 45.72 | 67.18 | 21.79 |
| | w/ AFT[4] (original) | 33.33 | 55.74 | 61.24 | **68.30** | 29.18 | **52.84** | **43.59** | **47.20** | **67.79** | **22.75** |
| GC | w/o AFT | 50.56 | 43.97 | 47.67 | **61.28** | 19.73 | 33.43 | 32.51 | 35.51 | **62.25** | 13.88 |
| | w/ AFT[4] (ours) | **79.09** | **68.78** | **64.72** | 59.38 | **29.93** | 35.48 | 33.39 | 37.01 | 62.21 | 14.63 |
| | w/ AFT[4] (original) | 77.84 | 67.75 | 63.48 | 58.88 | 29.19 | **35.64** | **33.96** | **37.84** | 62.03 | **15.15** |
| Grad-ECLIP | w/o AFT | 67.89 | 52.53 | 53.18 | 63.02 | 23.73 | – | – | – | – | – |
| | w/ AFT[4] (ours) | **85.39** | **68.74** | **72.19** | **69.70** | **35.07** | – | – | – | – | – |
| | w/ AFT[4] (original) | 84.39 | 68.12 | 70.58 | 68.99 | 34.28 | – | – | – | – | – |

Following the original protocol, we evaluate models on the intersection of the ImageNet validation set and ImageNet-Segmentation (Gao et al., 2023). As shown in Tab. 1, our retrained $AFT^4$ model closely matches the author-released checkpoint across both zero-shot classification and linear probing, confirming model-level reproducibility. However, our absolute metric values exhibit a uniform shift compared to those reported by Gong et al. (2025) (e.g., lower PG for Grad-CAM/Grad-ECLIP and a much higher baseline Mask IoU of 13.29 versus their 2.16). Because our retrained model and the released checkpoint yield nearly identical scores, this shift is clearly a protocol artifact caused by evaluation choices omitted in the original paper. Specifically, our pipeline applies a fixed top-20% intensity thresholding (topk = 0.2) to define foreground saliency pixels, which naturally establishes a rigid geometric baseline for accidental overlap with ImageNet masks that differs from the unstated thresholding heuristics of the original work. Similarly, variations in low-to-high resolution grid upsampling ($14 \times 14$ to $224 \times 224$) and our strict zero-tolerance pixel containment rule (versus an unstated hit-radius tolerance window) heavily influence argmax coordinates along complex mask boundaries. Crucially, because these protocol variations affect baseline and fine-tuned models uniformly, the relative scale of performance is thoroughly preserved: under AFT, Simple Gradient PG accuracy triples, PG-Energy increases by $\approx 50\%$, and Average Precision improves substantially, fully validating the core claims of Claim 1.

### 4.1.2 Concept Alignment (Claim 2)

To quantitatively evaluate the interpretability of the visual encoder, we reproduced a network dissection experiment following the CLIP-dissect (Oikarinen & Weng, 2023) methodology. The primary goal of this

analysis is to identify individual neurons within the visual representation that selectively respond to specific, human-understandable concepts.

**Experimental Setup.** In our implementation, we define these neurons as the individual activation channels of the [CLS] token extracted from the final layer of the Vision Transformer (ViT). Specifically, we utilize the Broden dataset as a probing ground. For each neuron $k$, we record its activation $A_k$ across the probing images and measure its alignment with a concept similarity array $S_c$, generated by a pre-trained CLIP model. A neuron is formally classified as a detector for a specific concept if it exhibits the highest alignment score with that concept among the entire candidate set.

While the authors mentioned using the 20,000 most common English words, they did not specify any filtering methods or provide the exact list of words used. To address this and ensure reproducibility, we utilised the `wordfreq` (Speer, 2022) library to extract the top 20,000 most frequent English words.

Additionally, while the original study primarily focused on a single alignment metric, our study expands this evaluation by benchmarking performance across three distinct similarity functions to ensure a more robust analysis:

- **Soft WPMI**: The default CLIP-dissect metric utilizes Soft Weighted Pointwise Mutual Information to highlight semantically specific concepts while penalizing overly frequent terms.

- **Cosine Similarity**: A standard linear correlation measure between the neuron's activation pattern and the CLIP concept similarity.

- **Cubed Cosine Similarity**: A non-linear variant that amplifies peak activations, allowing us to identify sparse detectors that respond only to highly specific visual incentives.

By evaluating all three metrics on a cleaned, verifiable concept set, we provide a more comprehensive view of how the AFT method influences the emergence of interpretable features compared to the baseline.

Table 2: Mean and median alignment scores for different AFT strengths.

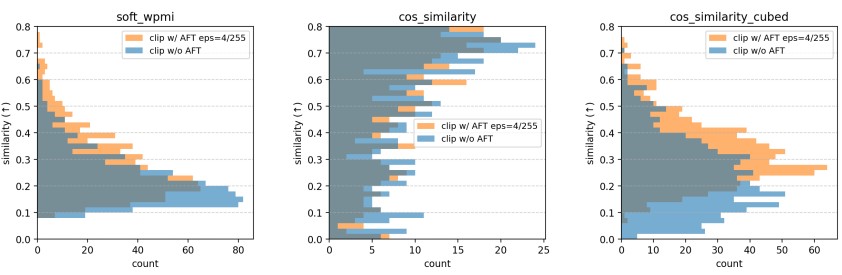

Figure 2: Distribution of neuron-concept alignment scores under three similarity metrics for CLIP with and without AFT.

|  | Mean | Median |
|---|---|---|
| w/o $AFT$ | | |
| Soft-WPMI | 0.2264 | 0.2023 |
| Cos | 0.2852 | 0.4437 |
| Cos-cubed | 0.2369 | 0.2205 |
| w/ $AFT^1$ | | |
| Soft-WPMI | 0.2610 | 0.2307 |
| Cos | 0.3046 | 0.4726 |
| Cos-cubed | 0.3051 | 0.2853 |
| w/ $AFT^4$ | | |
| Soft-WPMI | 0.2817 | 0.2558 |
| Cos | 0.3065 | 0.4996 |
| Cos-cubed | 0.3289 | 0.3124 |

**Results.** As shown in Figure 2 and Table 2, AFT consistently improves neuron-concept alignment across all three similarity metrics compared to the baseline. Both the mean and median scores increase monotonically with stronger AFT, indicating that AFT promotes the emergence of more semantically aligned and increasingly sparse concept detectors within the visual encoder.

We also apply the original Network Dissection protocol (Bau et al., 2017) to the final visual token of CLIP's image encoder (768 neurons, $14 \times 14$ spatial grid) using the Broden dataset. For each neuron, we estimate the top-0.5% activation threshold, upsample the binarized activation maps to input resolution, and compute

IoU with each concept's segmentation mask. A neuron counts as a detector when its best-matching concept exceeds IoU $\geq 0.04$.

Table 3: Network Dissection on the Broden dataset (768 neurons, IoU $\geq 0.04$ threshold).

| Model | Detectors | % neurons |
|---|---|---|
| CLIP w/o AFT | 115/768 | 15.0% |
| $AFT^1$ (ours) | 187/768 | 24.3% |
| $AFT^4$ (ours) | 263/768 | 34.2% |
| $AFT^4$ (original) | 258/768 | 33.6% |

$AFT^4$ more than doubles the number of concept detectors relative to the baseline, and our retrained $AFT^4$ closely matches the released checkpoint. However, the absolute counts differ substantially from the original paper (ours: 115→263; original: 42→159), likely due to unspecified details in their Network Dissection setup. We therefore view Claim 2 as reproduced in trend, but not quantitatively matched.

### 4.1.3 Transferability (Claim 3)

**Experimental Setup.** We evaluated the transferability of interpretability improvements to both out-of-distribution (OOD) domains and downstream classification tasks. Our experiments reproduced the settings of Figures 2(c,d) from the original paper.

For the OOD evaluation in Figure 1(c), we tested the models on four diverse datasets spanning different visual domains: ISIC 2024 (dermatology), Chest X-ray (radiology), Wildlife Animals (natural photography), and Kvasir-SEG (colonoscopy). For each dataset, we used domain-specific text prompts, such as "a dermoscopy image of melanoma" for melanoma labeled images in ISIC.

For downstream task evaluation (Figure 1(d)), we trained linear classification heads on frozen CLIP features. Specifically, we trained linear probes on COCO images for a 6-class classification task ("person", "car", "dog", "cat", "bird", "chair") using both the baseline CLIP encoder and the AFT encoder. We then generated Simple Gradient saliency maps for the resulting classifiers.

**Results.** Across all four OOD domains, the AFT model produced saliency maps with clearer and more localised focus on semantically relevant image regions compared to the baseline CLIP. This effect was particularly pronounced in medical imaging datasets (ISIC, Chest X-ray, and Kvasir-SEG), which represent a substantial distribution shift from ImageNet.

Importantly, this improved gradient structure also persisted in downstream classification. The linear probe trained on frozen AFT features generated more structured and interpretable saliency maps than the probe trained on baseline CLIP features, with gradients remaining focused on meaningful object regions. This indicates that the benefits of adversarial fine-tuning transfer not only across domains, but also across tasks and model usage settings.

### 4.1.4 Accuracy Trade-off (Claim 4)

The current experiment evaluates whether the gains in interpretability and robustness come at a high cost to the model's standard performance.

**Experimental setup.** We compare standard CLIP, the checkpoint provided by the original paper ($AFT^4$ original), and our two models with $\varepsilon = 1$ and $\varepsilon = 4$, respectively. We evaluate the models' zero-shot accuracy on 1,000 random samples from both in-distribution datasets (similar to the training set) and out-of-distribution datasets. We also evaluate them under two attacks from AutoAttack (Croce & Hein, 2020a). These attacks consist of APGD with Cross-Entropy (CE) loss followed by APGD with targeted Difference Logits Ratio (DLR) loss toward the nearest incorrect class. We use 100 iterations for the attacks with maximum $L_\infty$ perturbations of 2/255 and 4/255

Table 4: Adversarial robustness and accuracy results. (ours) denotes retrained models; (original) denotes original checkpoints provided by authors.

| $L_\infty$ | Model | CIFAR-10 | STL-10 | EuroSAT | FGVC Air. | Imagenette | Average |
|---|---|---|---|---|---|---|---|
| clean | CLIP | **89.4** | **98.1** | **40.50** | **22.0** | **13.5** | **52.70** |
| | AFT[1] (ours) | 86.4 | 97.7 | 9.5 | 21.4 | 12.0 | 45.40 |
| | AFT[4] (ours) | 66.2 | 91.4 | 10.5 | 12.2 | 12.2 | 38.50 |
| | AFT[4] (original) | 63.2 | 90.4 | 11.90 | 12.0 | 12.2 | 37.94 |
| 2/255 | CLIP | 0.0 | 0.0 | 0.0 | 0.0 | 0.0 | 0.00 |
| | AFT[1] (ours) | 20.3 | 56.1 | 0.6 | 1.8 | 0.6 | 15.88 |
| | AFT[4] (ours) | **44.9** | **79.6** | 10.1 | 3.2 | 5.2 | **28.60** |
| | AFT[4] (original) | 41.6 | 77.2 | **11.90** | **3.3** | **5.8** | 27.96 |
| 4/255 | CLIP | 0.0 | 0.0 | 0.0 | 0.0 | 0.0 | 0.00 |
| | AFT[1] (ours) | 0.2 | 6.1 | 0.0 | 0.0 | 0.0 | 1.26 |
| | AFT[4] (ours) | **24.6** | **58.4** | 10.0 | 1.1 | 2.1 | **19.24** |
| | AFT[4] (original) | 22.4 | 57.4 | **11.70** | **1.2** | **2.3** | 19.00 |

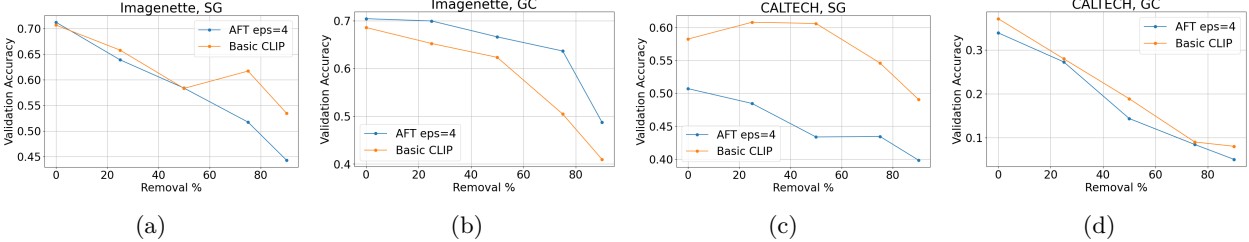

(a)          (b)          (c)          (d)

Figure 3: ROAR results on Imagenette and Caltech datasets with SG and GC saliency maps

**Reproducibility.** The absolute numerical results can vary significantly depending on the specific subset of each dataset and the way attacks are assembled. However, the overall trend of zero robust accuracy for basic CLIP under attack and robust behaviour for fine-tuned models remains the same. Additionally, the provided checkpoint weights and our model trained with the same setup result in similar accuracies.

### 4.1.5 Remove and Retrain

To prove that saliency maps after AFT highlight the task-related regions of the input image, the authors of the original paper use the Remove and Retrain (ROAR) method (Hooker et al., 2019). ROAR measures how much the actual information content of the dataset decreases by forcing a new model to learn from the modified masked data.

**Experimental setup.** First, we generate static saliency maps for the entire training and validation sets of Caltech256 (Griffin et al., 2022) and Imagenette (Howard, 2019) using a reference AFT[4] model. We then construct multiple masked versions of the dataset by removing the top $k \in \{25\%, 50\%, 75\%, 95\%\}$ most salient pixels, as identified by the saliency method. For each masking level, a ResNet-50 visual backbone is initialised and trained from scratch on the corresponding masked training set. We repeat each experiment three times with random initialization to decrease the noise in the results. The model is subsequently evaluated via linear probing on the matching masked validation set.

This protocol ensures that any observed performance degradation can be attributed to the removal of semantically relevant information.

**Results.** As illustrated in Fig. 3, three out of the four experimental settings (a, c, d) demonstrate that classification accuracy degrades significantly more rapidly when using saliency maps from AFT-enhanced models compared to standard CLIP. This suggests that AFT produces saliency maps that are more effectively concentrated on task-relevant features. However, the evaluation on Imagenette using Grad-CAM (b) does not follow this trend.

Consequently, ROAR provides only partial support for Claim 1. While the expected trend holds in the majority of settings, the Imagenette/Grad-CAM failure case and the quantitative mismatch with the original plots weaken the evidence for a perfect numerical reproduction. This mismatch likely stems from the fact that the original work does not explicitly detail the exact experimental configurations used for the ROAR benchmarks.

## 4.2 Extensions to the original paper

### 4.2.1 Cross-Modal Retrieval Robustness on Flickr30k

Table 5: Cross-modal retrieval results (Recall@k, %) on the Flickr30k dataset under different perturbation budgets. $AFT^i$ denotes models trained with perturbation $\epsilon = i/255$. (ours) denotes retrained models; (original) denotes original checkpoints provided by authors.

| $L_\infty$ | Vision Encoder | T→I (%) | | I→T (%) | |
|---|---|---|---|---|---|
| | | R@1 | R@5 | R@1 | R@5 |
| clean | CLIP baseline | 24.95 | 45.37 | **44.41** | **68.61** |
| | $AFT^1$ (*ours*) | **27.20** | **48.52** | 43.02 | 67.09 |
| | $AFT^4$ (*original*) | 19.53 | 37.50 | 26.38 | 47.54 |
| | $AFT^4$ (*ours*) | 19.53 | 37.46 | 26.33 | 47.78 |
| 2/255 | CLIP baseline | 1.83 | 4.74 | 2.25 | 5.66 |
| | $AFT^1$ (*ours*) | **18.59** | **35.95** | **30.72** | **52.05** |
| | $AFT^4$ (*original*) | 17.88 | 34.73 | 24.07 | 43.66 |
| | $AFT^4$ (*ours*) | 17.84 | 34.69 | 24.20 | 44.05 |
| 4/255 | CLIP baseline | 0.18 | 0.57 | 0.22 | 0.72 |
| | $AFT^1$ (*ours*) | 7.80 | 17.15 | 13.60 | 26.43 |
| | $AFT^4$ (*original*) | 13.96 | 27.71 | 18.36 | 34.61 |
| | $AFT^4$ (*ours*) | **14.07** | **27.89** | **18.79** | **35.51** |

**Experimental Setup.** To extend the evaluation beyond the benchmarks considered in the original paper, we evaluate cross-modal retrieval robustness on the Flickr30k dataset.

Performance is measured using Recall@1 (R@1) and Recall@5 (R@5) under clean conditions as well as under adversarial perturbations with $\epsilon \in \{2/255, 4/255\}$, consistent with the models studied in the original work.

We evaluate the original CLIP model alongside models trained with AFT. For $AFT^4$, we report results for both the author-provided checkpoint (*original*) and an independently retrained checkpoint (*ours*), enabling an assessment of reproducibility. Table 5 summarises the retrieval performance.

**Clean Performance ($\epsilon = 0$).** Under clean conditions, our retrained $AFT^1$ model improves text-to-image retrieval performance compared to the CLIP baseline, achieving higher Recall@k scores. This suggests that adversarial fine-tuning with a small perturbation budget can act as a form of regularisation, improving semantic alignment between image and text embeddings without degrading clean performance.

**Adversarial Robustness ($\epsilon = 2/255, 4/255$).** Under adversarial perturbations, the retrieval performance of the standard CLIP model rapidly degrades, collapsing to near-zero recall even at moderate perturbation levels ($\epsilon = 2/255$). In contrast, AFT-trained models retain substantial retrieval capability across both threat models. At $\epsilon = 2/255$, models trained with smaller perturbation budgets achieve a favourable balance between robustness and accuracy, while under stronger attacks ($\epsilon = 4/255$), $AFT^4$ models significantly outperform all other configurations.

**Reproducibility.** The close agreement between the performance of our retrained $AFT^4$ model and the publicly available checkpoint demonstrates that the robustness gains reported in the original paper are reproducible.

### 4.2.2 Saliency-Guided Regularization

**Experimental setup.** We evaluate the fine-tuning strategy described in Section 3.3 on the COCO 2017 validation set using Simple Gradient, Grad-CAM, and Grad-ECLIP saliency maps, sweeping the saliency loss weight $w_s \in \{0.2, 0.4, 0.6\}$. We report the same five localisation metrics used in Section 4 and evaluate adversarial robustness on five zero-shot classification benchmarks, following the same protocol as in Section 4.

**Saliency map quality.** Table 6 reports localisation metrics on ImageNet-S across the four settings ($w_s = 0$, corresponding to the COCO-finetuned AFT[4] model without saliency loss, and $w_s \in \{0.2, 0.4, 0.6\}$). For Simple Gradient, every metric improves monotonically with $w_s$, with the strongest gain observed for the Point Game metric ($+40\%$ relative at $w_s = 0.6$). Grad-CAM metrics remain largely unchanged across all configurations, consistent with our earlier finding that Grad-CAM is unreliable for CLIP-based models.

**Transfer to Grad-ECLIP.** Notably, the saliency loss also improves Grad-ECLIP despite never optimising it directly: all five localisation metrics improve monotonically with $w_s$. This indicates that the saliency loss does not merely overfit the Simple Gradient pathway but induces a structural change in how the encoder distributes gradient information across the input. This cross-method transfer provides some of the strongest evidence that the saliency regularisation improves the encoder's internal representations. A qualitative counterpart to Fig. 1, showing the same image–prompt pairs under saliency-guided regularisation at $w_s = 0.6$, is provided in Appendix C; the accompanying repository additionally contains the equivalent figures for $w_s \in \{0.2, 0.4\}$.

Table 6: Localisation metrics on the ImageNet-S validation set for three saliency methods. *Baseline*: standard adversarial fine-tuning without saliency loss ($w_s = 0$). Remaining rows apply saliency fine-tuning over the converged robust checkpoint with different values of the saliency loss weight $w_s$.

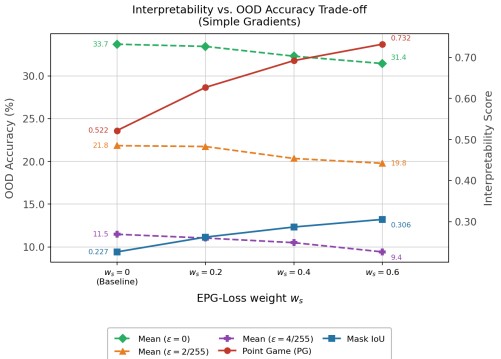

Figure 4: Interpretability versus zero-shot OOD accuracy trade-off as a function of the EPG-loss weight $w_s$, using Simple Gradient saliency maps. Solid lines: interpretability metrics (right axis). Dashed lines: mean zero-shot accuracy across five benchmarks under $\ell_\infty$ perturbations of $\varepsilon \in \{0, 2/255, 4/255\}$ (left axis).

| Method | Sal. | PG↑ | E-PG↑ | AP↑ | P.Acc.↑ | IoU↑ |
|---|---|---|---|---|---|---|
| | SG | 0.5218 | 0.4307 | 0.4688 | 0.6776 | 0.2265 |
| $w_s = 0$ | GC | 0.1688 | 0.1966 | **0.2612** | **0.5525** | 0.0553 |
| | Grad-ECLIP | 0.4470 | 0.3570 | 0.4201 | 0.6422 | 0.1822 |
| | SG | 0.6269 | 0.4845 | 0.5289 | 0.6990 | 0.2625 |
| $w_s = 0.2$ | GC | 0.1653 | 0.1942 | 0.2552 | 0.5487 | 0.0520 |
| | Grad-ECLIP | 0.5149 | 0.3789 | 0.4412 | 0.6507 | 0.1960 |
| | SG | 0.6922 | 0.5256 | 0.5709 | 0.7129 | 0.2871 |
| $w_s = 0.4$ | GC | 0.1736 | 0.2000 | 0.2543 | 0.5499 | 0.0529 |
| | Grad-ECLIP | 0.5695 | 0.3993 | 0.4591 | 0.6571 | 0.2075 |
| | SG | **0.7321** | **0.5606** | **0.6034** | **0.7231** | **0.3056** |
| $w_s = 0.6$ | GC | **0.1941** | **0.2078** | 0.2595 | 0.5520 | **0.0585** |
| | Grad-ECLIP | **0.6095** | **0.4145** | **0.4708** | **0.6611** | **0.2148** |

**Interpretability-accuracy trade-off.** Figure 4 jointly visualises the evolution of interpretability and zero-shot accuracy as $w_s$ increases. The two interpretability metrics (Point Game and Mask IoU, solid lines) rise steadily with $w_s$, while mean zero-shot accuracy under clean conditions and under adversarial attack at $\varepsilon \in \{2/255, 4/255\}$ (dashed lines) decreases gracefully. The accuracy drop is marginal at $w_s = 0.2$ and remains moderate even at $w_s = 0.6$, while the interpretability gains at those settings are substantial. The $w_s$ sweep thus reveals a favourable trade-off curve: a small sacrifice in zero-shot accuracy buys a disproportionately large improvement in saliency alignment, with the choice of $w_s$ determined by whether the downstream application prioritises spatial faithfulness of attributions or robustness under adversarial perturbation.

**Effect on concept alignment.** To test whether the saliency extension affects the encoder's internal representations beyond gradient structure, we repeat the Network Dissection evaluation from Section 4.1.2 for all EPG-fine-tuned models. As shown in Table 7, the number of concept detectors stays high across

Table 7: Network Dissection on the Broden dataset. Effect of saliency-guided regularisation with varying $w_s$ on concept detector count.

| Model | Detectors | % neurons |
|---|---|---|
| $\text{AFT}^4$ + EPG ($w_s = 0$) | 295/768 | 38.4% |
| $\text{AFT}^4$ + EPG ($w_s = 0.2$) | 277/768 | 36.1% |
| $\text{AFT}^4$ + EPG ($w_s = 0.4$) | 285/768 | 37.1% |
| $\text{AFT}^4$ + EPG ($w_s = 0.6$) | 298/768 | 38.8% |

all $w_s$ values (277-298). This indicates that the EPG loss leaves neuron-concept alignment unchanged: it modifies where gradient mass is concentrated, rather than which concepts individual neurons detect. Thus, the localization gains from saliency supervision do not come with an observable loss in the encoder's semantic structure.

**Results.** The saliency loss yields substantial improvements in foreground alignment for Simple Gradient, the attribution method it directly optimises, with partial transfer to Grad-ECLIP and little effect on Grad-CAM. We note that the Energy PG metric is closely related to the training objective, so improvements on this metric are partly expected by construction. The concurrent improvements in Point Game, which depends only on the argmax pixel, and Mask IoU, which measures thresholded region overlap, provide more independent evidence that the model genuinely attends to foreground regions rather than merely redistributing gradient mass. Overall, explicit saliency supervision delivers a favourable interpretability-accuracy trade-off and is complementary to the implicit gradient regularisation provided by AFT.

## 5 Discussion and Conclusion

### 5.1 Main Findings and Reproducibility

This reproduction study provides directional support for the main conclusions of Gong et al. (2025). Across saliency localisation, neuron–concept alignment, transferability, and adversarial robustness, we consistently observe the same qualitative trends as the original paper: AFT improves gradient-based interpretability, increases concept alignment, generalises to out-of-distribution domains and downstream tasks, and provides substantial robustness at the cost of reduced clean accuracy.

However, several absolute metric values differ from the original paper, especially for localisation metrics, Network Dissection counts, and ROAR curves. Since our retrained checkpoints closely match the released checkpoints, we attribute these discrepancies primarily to unspecified evaluation protocol details rather than model-level differences. We therefore view our reproduction as confirming the direction and relative scale of the reported effects, but not their exact numerical values.

Our saliency-guided regularisation extension further shows that explicit foreground-mask supervision can substantially improve localisation metrics while preserving most of the adversarial robustness. This suggests that the implicit gradient regularisation induced by AFT can be effectively complemented with direct spatial supervision. In Appendix E, we show that the EPG loss can be interpreted as a background-restricted adversarial objective, placing it within the same duality framework that underpins AFT.

Finally, we provide a full accounting of the computational resources and environmental impact of this study in Appendix D.

### 5.2 Practical Challenges and Limitations

While the core claims of Gong et al. (2025) are reproducible, the process was hindered by incomplete evaluation scripts in the original repository, requiring a ground-up reimplementation of the localization and robustness protocols. Methodologically, we observed that Grad-CAM visualizations for CLIP-style encoders exhibit instability, suggesting that gradient-based interpretability in these architectures requires further refinement or alternative saliency methods.

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

# A    Datasets

Table 8: Datasets used in our reproduction study.

| Dataset | Usage | #images |
|---------|-------|---------|
| ImageNet 2012 | Adversarial fine-tuning | 1.28M |
| Flickr30k | Retrieval evaluation | 31K |
| ISIC 2024 | OOD saliency evaluation | 401K |
| Chest X-ray | OOD saliency evaluation | 5.9K |
| Wildlife Animals | OOD saliency evaluation | 5.2K |
| Kvasir-SEG | OOD saliency evaluation | 1K |
| Broden | Neural-concept alignment | 63K |
| **Zero-shot Evaluation Datasets (Validation/Test Splits)** | | |
| CIFAR-10 | Zero-shot evaluation | 10K |
| STL-10 | Zero-shot evaluation | 8K |
| EuroSAT | Zero-shot evaluation | 5.4K |
| FGVC Aircraft | Zero-shot evaluation | 3.3K |
| Imagenette | Zero-shot evaluation, ROAR analysis | 3.9K |

# B    Adversarial Fine-Tuning Formulation

The original paper formulates AFT as a minimax optimisation problem that enforces robustness while implicitly regularising input gradients. We present the key equations following Gong et al. (2025), noting the correspondence to their original formulation.

Given an image $x$, its image embedding $I_x = f_\theta(x)$ (unit-normalized), and corresponding text embedding $T_x$ (unit-normalized), the supervised AFT objective is:

$$\min_\theta \mathbb{E}_{x \sim \mathcal{D}_{\text{train}}} \max_{\delta_x} \left[ \frac{1}{2} \left( T_x^\top \mathbb{E}_{z \sim \mathcal{N}(0,\sigma^2 I)}[f_\theta(x + z + \delta_x)] - T_x^\top I_x \right)^2 - h(\delta_x) \right] \tag{3}$$

where $\delta_x$ is the adversarial perturbation, $z$ is Gaussian smoothing noise that improves optimization stability, and $h(\cdot)$ is a regularization term that constrains the perturbation magnitude. This formulation corresponds to Equation (1) in Gong et al. (2025).

Using a first-order Taylor expansion around $\delta_x = 0$, the inner maximization can be approximated as:

$$\max_{\delta_x} \left[ \delta_x^\top \omega_x \nabla_x \mathbb{E}_{z \sim \mathcal{N}(0,\sigma^2 I)}[T_x^\top f_\theta(x + z)] - h(\delta_x) \right] \tag{4}$$

where $\omega_x = |T_x^\top \mathbb{E}_z[f_\theta(x + z)] - T_x^\top I_x|$. The gradient term $\nabla_x \mathbb{E}_z[T_x^\top f_\theta(x + z)]$ is precisely the SmoothGrad saliency map (Smilkov et al., 2017), revealing the connection between AFT and gradient regularisation. This approximation corresponds to Equation (2) in the original paper.

Through Fenchel duality, the optimisation can be reformulated as:

$$\min_\theta \mathbb{E}_{x \sim \mathcal{D}_{\text{train}}} \left[ m_x(0) + h^* \left( \omega_x \nabla_x \mathbb{E}_z[T_x^\top f_\theta(x + z)] \right) \right] \tag{5}$$

where $h^*(\cdot)$ is the Fenchel conjugate of $h(\cdot)$ and $m_x(\delta_x)$ denotes the squared similarity loss. This reformulation, derived in Section 3.1 of Gong et al. (2025), shows that AFT implicitly applies a penalty $h^*(\cdot)$ directly to the saliency map, enforcing sparsity in the gradient-based explanations.

The paper uses a smooth $\ell_1$ (Huber) norm (Huber, 1964) for regularization, with dual form:

$$h^*(u) = \varepsilon \sum_i H_\eta(u^{(i)}), \quad \text{where } H_\eta(u^{(i)}) = \begin{cases} \frac{1}{2\eta}(u^{(i)})^2 & \text{if } |u^{(i)}| \leq \eta \\ |u^{(i)}| - \frac{\eta}{2} & \text{otherwise} \end{cases} \tag{6}$$

where $\varepsilon$ controls the perturbation bound and $\eta$ is a smoothness parameter. This corresponds to Equation (4) in the original paper. The piecewise structure of the Huber function applies a quadratic penalty for small gradient values while applying a linear penalty (promoting sparsity) for large values, effectively encouraging the network to concentrate attribution on a few salient regions rather than distributing it across the entire input. This leads to clearer and more interpretable gradient-based explanations.

## C   Qualitative Effect of Saliency-Guided Regularisation

Figure 5 complements the quantitative results of Section 4.2.2 with a qualitative comparison that mirrors the layout of Figure 1. For the same image–prompt pairs used in the main text, we show Simple Gradient and Grad-ECLIP saliency maps produced by the AFT[4] backbone further fine-tuned with the saliency-guided regularisation objective described in Section 3.3 at $w_s = 0.6$.

We generated the same figure for every sweep value $w_s \in \{0.2, 0.4, 0.6\}$. Across the sweep, each increment of $w_s$ yields a marginal visual improvement over the previous setting: saliency mass becomes progressively more concentrated on the referenced object, and spurious activations outside the regions of interest are progressively weakened. This trend holds for both Simple Gradient and Grad-ECLIP, the $w_s = 0.6$ setting produces the visually cleanest maps, with the fewest off-object artifacts for both attribution methods. Because the gains between consecutive sweep values are small, we include only the $w_s = 0.6$ figure in the paper and defer the remaining variants to the accompanying code repository.

These qualitative observations are consistent with the monotonic trend in the localisation metrics reported in Table 6 and reinforce our reading that saliency-guided regularisation induces a structural change in how the encoder distributes gradient information, rather than merely rescaling it.

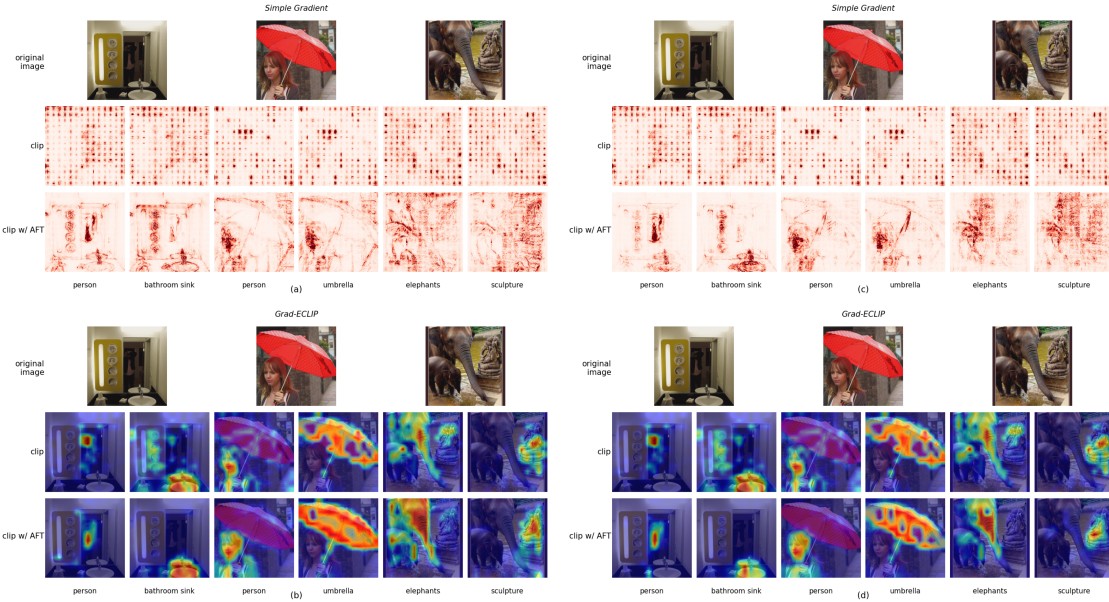

Figure 5: Saliency map comparison between CLIP (middle row) and CLIP with AFT (bottom row) for the original AFT[4] backbone (left column, as in Figure 1) and the AFT[4] backbone with saliency supervision (right column). Images (a) and (c) are Simple Gradient maps for different text prompts, images (b) and (d) are Grad-ECLIP visualisations for the same images. Saliency mass is visibly more concentrated on the referenced object and contains fewer off-object artifacts than the corresponding AFT[4] maps in Figure 1.

## D  Environmental Impact

We estimated the carbon footprint of our experiments using the CodeCarbon (Courty et al., 2024) framework, which provides automated tracking of hardware-level energy consumption and converts it into carbon emissions. Specifically, total emissions were computed as:

$$CO_2e = CI \cdot PUE \cdot E,$$

where $CI$ denotes the carbon intensity of electricity generation, $PUE$ is the power usage effectiveness of the computing infrastructure, and $E$ represents the total electrical energy consumed by the CPU, GPU, and memory during the experiments.

All reported values correspond to the computational resources required to train the main AFT[4] model from scratch, following the experimental protocol described in 3.2. Over a total runtime of approximately **34.6 hours**, the experiments consumed **28.7 kWh** of electricity.

Using the average carbon intensity for the country where experiments were conducted in 2024 and assuming a PUE of 1.0 (as reported by CodeCarbon), this corresponds to an estimated **7.7 kg of CO₂e emissions**. Importantly, these estimates are fully reproducible, as they are directly derived from the raw CodeCarbon logs.

Beyond the initial training, we conducted additional experiments including backbone pre-training on the COCO dataset and four subsequent fine-tuning with saliency-guided regularization runs using 2 NVIDIA A100-40GB GPUs. The COCO pre-training stage (ViT-B-16, 5,000 steps) required approximately **5.9 hours** and consumed **4.97 kWh**, resulting in **1.33 kg CO₂e**. The subsequent fine-tuning runs added a cumulative **11.9 hours** of compute and **9.89 kWh** of energy, emitting an additional **2.65 kg CO₂e**. This complete pipeline (pre-training and four fine-tuning runs) accounts for approximately **14.86 kWh** and **3.98 kg CO₂e**. Notably, a single fine-tuning run emitted approximately **0.66 kg CO₂e**, representing roughly half the footprint of the backbone pre-training stage.

## E  Theoretical Properties of the Saliency-Guided Objective

While the saliency-guided regularization in Section 3.3 is introduced on empirical grounds, it admits two complementary readings that clarify its optimization behaviour and its relationship to the adversarial framework of Gong et al. (2025). Throughout this section we write $s(x;\theta) = I_{\mathrm{orig}}^{\top} f_\theta(x)$ for the feature consistency score and $g_x = \nabla_x s(x;\theta)$ for its input gradient. The unnormalized Simple Gradient attribution at pixel $p$ is $S_p = \sum_c |g_{x,p}^{(c)}|$, and we denote by $S = \{S_p\}_{p\in\Omega}$ the full spatial map, so that $\|S\|_1 = \sum_{p\in\Omega} S_p$ and $\mathcal{L}_{\mathrm{EPG}} = 1 - \|S\|_1^{-1} \sum_{p\in M} S_p$.

**The EPG gradient is a mixed input-parameter Hessian.** Differentiating $\mathcal{L}_{\mathrm{EPG}}$ through the absolute value and the normalisation yields

$$\nabla_\theta \mathcal{L}_{\mathrm{EPG}} \;=\; \sum_{p\in\Omega} C_p \,\mathrm{sgn}(g_{x,p}) \cdot \frac{\partial^2 s(x;\theta)}{\partial\theta\,\partial x_p}, \tag{7}$$

where each $C_p = C_p(\theta, x, M)$ is a scalar weight that depends on the normaliser $\|S\|_1$ and on whether $p$ lies inside $M$; its sign encodes the direction in which pixel $p$ should move (saliency mass increases on $M$ and decreases on the background $M^c = \Omega \setminus M$). The update is therefore driven by the *mixed input-parameter Hessian $\partial^2 s/\partial\theta\,\partial x$*: EPG does not adjust first-order activations, it reshapes how parameter sensitivity is distributed across the input grid, pulling it onto $M$ and away from $M^c$. In practice we treat $\|S\|_1$ as a constant during the backward pass, so that $\nabla_\theta \mathcal{L}_{\mathrm{EPG}}$ is driven only by the redistribution of saliency mass between $M$ and $M^c$ and not by changes in its total magnitude.

This second-order structure also explains why the joint objective cannot be optimized from a CLIP initialization. The per-sample estimate of Eq. 7 is a sum of Hessian entries weighted by $\mathrm{sgn}(g_{x,p})$. When $g_x$ is dense and near-uniform, as it is for pre-trained CLIP, these signs are effectively random across pixels, and the

estimate has high variance. Once the implicit Huber-dual penalty $h^\star(\omega_x g_x)$ induced by AFT has made $g_x$ sparse, the sum collapses onto the few active pixels, and the EPG gradient becomes a well-posed reshaping signal. This is why saliency-guided fine-tuning must follow AFT rather than run alongside it.

**EPG as a background-restricted adversarial objective.** The Hessian reading in Eq. 7 makes precise *how* EPG reshapes the encoder; a second, adversarial reading makes precise *what* it is asking the encoder to be invariant to, and mirrors the Taylor-duality argument that Gong et al. (2025) use to connect their Eq. (1) and Eq. (2). Let $\mathcal{A}_{M^c}(\varepsilon) = \{\delta : \|\delta\|_\infty \leq \varepsilon, \ \delta_p = 0 \ \forall p \in M\}$ denote the set of $\ell_\infty$-bounded perturbations supported on the background $M^c = \Omega \setminus M$, and consider the worst-case change of the similarity score under such perturbations,

$$\Phi_M(x;\theta) \;=\; \max_{\delta \in \mathcal{A}_{M^c}(\varepsilon)} \big| s(x + \delta; \theta) - s(x; \theta) \big|. \tag{8}$$

**Proposition 1.** *To first order in $\varepsilon$,*

$$\Phi_M(x;\theta) \;=\; \varepsilon \sum_{p \notin M} \big| g_{x,p} \big| \;+\; O(\varepsilon^2). \tag{9}$$

*Consequently, up to the normalizing constant $\|S\|_1$ and the perturbation budget $\varepsilon$, minimizing $\Phi_M$ over $\theta$ is equivalent to leading order to minimizing $\mathcal{L}_{EPG}$.*

*Proof.* Taylor-expanding $s$ around $x$,

$$s(x + \delta; \theta) - s(x; \theta) = \langle g_x, \delta \rangle + O(\|\delta\|^2).$$

Since $\delta_p = 0$ for $p \in M$, the inner product reduces to $\sum_{p \notin M} \langle g_{x,p}, \delta_p \rangle$. Each term is independently maximised over the $\ell_\infty$ ball by $\delta_p = \varepsilon \operatorname{sgn}(g_{x,p})$, yielding $\Phi_M(x;\theta) = \varepsilon \sum_{p \notin M} |g_{x,p}| + O(\varepsilon^2)$. Dividing by $\|S\|_1$ recovers $\varepsilon \cdot \mathcal{L}_{\mathrm{EPG}}$. $\square$

Proposition 1 gives our explicit spatial supervision an implicit adversarial reading: penalizing background saliency mass is, to leading order, equivalent to training the encoder to be robust against an adversary permitted to perturb only the background of the input. This recovers the same Taylor-duality structure that underpins the original AFT formulation, but applied to a spatially-restricted perturbation set rather than an unrestricted $\ell_\infty$ ball. Viewed jointly, the two readings describe the same objective from complementary angles: Eq. 7 characterises the optimisation signal EPG sends into parameter space, while Eq. 9 characterises the invariance it induces in input space, and both reduce to statements about how the encoder distributes gradient mass between $M$ and $M^c$.

