# OpenReview forum: "[Re] Boosting the Visual Interpretability of CLIP via Adversarial Fine-Tuning"
_TMLR — Under review for TMLR_

### Review · Reviewer_it6u · 2026-06-09

**Summary Of Contributions:**

This paper is a reproducibility study of Gong et al. (2025), which proposed adversarial fine-tuning (AFT) with norm regularization to improve the visual interpretability of CLIP's image encoder. The authors attempt to independently verify four central claims: that AFT improves saliency map quality (Simple Gradient and Grad-CAM), increases neuron-concept alignment, transfers to out-of-distribution domains and downstream tasks, and maintains reasonable zero-shot accuracy. Beyond reproduction, the authors contribute a novel saliency-guided regularization extension using an Energy Pointing Game (EPG) loss that directly supervises gradient-based attribution maps against COCO instance segmentation masks. They also release a restructured codebase with full evaluation protocols not present in the original repository.Key strengths include the thorough and honest treatment of evaluation discrepancies (correctly attributing metric gaps to underspecified protocols rather than model failures), the cross-method transfer finding (EPG loss improves Grad-ECLIP despite not optimizing it directly), the theoretically grounded appendix connecting EPG to adversarial robustness via Taylor duality, and the addition of retrieval and ROAR evaluations. A key weakness is that Grad-CAM results remain unreliable and inconclusive throughout, and the ROAR analysis provides only partial support for the claims.

**Additional Comments:**

The paper is well-written and the experimental organization is clear. The honest accounting of where reproduction succeeded and where it didn't — rather than inflating agreement with the original — is exactly what the community needs from reproducibility studies. The decision to implement missing evaluation code from scratch and release it is a genuine service to researchers working in this space.
One minor note: the abstract states the EPG extension shows improvement "with only a modest reduction in adversarial robustness," which slightly understates the result. Figure 4 actually shows the clean accuracy drop is quite small at ws = 0.2 and only moderate at ws = 0.6, while interpretability gains are substantial at both settings. The framing in the abstract could be made more precise.

**Audience:**

Yes

**Audience Explanation:**

Reproducibility studies of highly cited foundation model work are valuable to the ML community, especially when the original paper leaves evaluation protocols underspecified. CLIP is a widely used backbone, and the question of whether adversarial fine-tuning reliably improves interpretability — and whether those improvements transfer across domains and tasks — is directly relevant to practitioners building on CLIP for downstream applications in medical imaging, retrieval, and visual grounding.
Beyond the reproduction, the EPG loss extension is a concrete and lightweight contribution. The finding that explicit foreground supervision induces a favorable interpretability-accuracy tradeoff, and that this transfers to a saliency method not directly optimized, is a meaningful scientific result that could inform future work on interpretability-aware training.

**Broader Impact Concerns:**

No significant concerns. The work improves interpretability of vision-language models, which has positive implications for trust, auditability, and deployment in sensitive domains (medical imaging is even demonstrated as an OOD setting). The adversarial fine-tuning procedure does not introduce capabilities that raise safety concerns, and the carbon footprint (roughly 7.7 kg CO2e for the main training run) is modest and transparently reported. No broader impact statement is required beyond what is already covered by the environmental impact appendix.

**Claims And Evidence:**

Yes

**Claims Explanation:**

The core reproducibility claims are well-supported. The authors retrain models from scratch and show their checkpoints match the author-released weights closely across all major metrics (Table 1, Table 4, Table 5), providing strong evidence of model-level reproducibility. Where absolute numbers diverge from the original paper — notably in Pointing Game, Network Dissection counts, and ROAR curves — the authors give transparent and technically credible explanations (unspecified binarization thresholds, upsampling strategies, hit-radius tolerances, IoU thresholds). The observation that their retrained checkpoint and the released checkpoint give near-identical scores, yet both differ from the reported numbers, makes a strong case that the discrepancy is a protocol artifact rather than a modeling failure. This is an exemplary handling of a common problem in reproducibility work.For the extension, the EPG loss improvements are documented quantitatively in Table 6 and Figure 4 across five metrics and three saliency methods. The cross-method transfer to Grad-ECLIP (never directly optimized) is particularly compelling evidence that the loss induces a structural change in the encoder rather than a superficial gradient rescaling. The theoretical analysis in Appendix E connecting EPG to a background-restricted adversarial objective is rigorous and well-presented.Some caveats: the ROAR analysis in one of four conditions (Imagenette/Grad-CAM) does not follow the expected trend, and Grad-CAM is treated as inconclusive throughout. These are acknowledged honestly rather than swept under the rug, which is appropriate. The paper would benefit from a clearer discussion of why the Grad-CAM failure is expected rather than concerning.

**Requested Changes:**

The Grad-CAM unreliability is noted repeatedly but never explained mechanistically. Given that one ROAR result (Imagenette/Grad-CAM) fails to reproduce the expected trend, the paper should either provide a hypothesis grounded in CLIP's architecture for why Grad-CAM is unreliable here, or include a brief experiment (e.g., on a standard CNN where Grad-CAM is known to work) that isolates the source of the instability. As currently written, treating Grad-CAM results as uniformly "inconclusive" weakens the paper's overall evidence base without sufficient justification.
The ROAR analysis (Section 4.1.5) is described but the experimental setup omits key details: what masking fill value is used (mean pixel, zero, noise)? Is the ResNet-50 trained with standard data augmentation? These choices substantially affect the absolute results and should be documented.
Table 1 would benefit from a brief note directly in the caption explaining the source of the numerical discrepancy with Gong et al. (2025), so readers can interpret the table without hunting through the text.
The cross-modal retrieval results (Table 5) show AFT1 improves clean text-to-image recall over the CLIP baseline. This is a somewhat surprising finding worth a sentence or two of discussion — is this a known regularization effect of mild adversarial training on contrastive embeddings?
Appendix E is technically strong but could be made more accessible with a short intuitive summary at the start of the main text (Section 3.3) pointing readers to what the appendix establishes. The theoretical connection between EPG and adversarial robustness is one of the paper's more novel contributions and currently undersold.
The paper mentions computational cost and carbon emissions in Appendix D, which is commendable. A one-sentence summary of this in the main text would make it easier for readers to assess the computational accessibility of the method.

---

### Review · Reviewer_Rzen · 2026-06-17

**Summary Of Contributions:**

This submission is a reproducibility study of Gong et al. (2025), which proposed adversarial fine-tuning (AFT) to improve the visual interpretability of CLIP. The paper provides useful independent evidence that AFT improves gradient-based localization trends and robustness, and it adds a supervised saliency-guided regularization extension based on an Energy Pointing Game loss. Overall, the study is valuable and mostly careful, but the conclusions are sometimes stronger than the evidence: several claims are reproduced only qualitatively or directionally, important protocol details remain ambiguous, and the proposed extension is not yet evaluated strongly enough to establish improved faithfulness beyond foreground localization.

**Audience:**

Yes

**Audience Explanation:**

The paper should interest researchers working on CLIP interpretability and adversarial robustness.

**Broader Impact Concerns:**

I do not see major negative broader impact concerns.

**Claims And Evidence:**

No

**Claims Explanation:**

- The claim that AFT improves saliency localization trends is reasonably supported, especially for Simple Gradient and Grad-ECLIP.
- The concept-alignment and transferability claims are suggestive but not fully convincing because they depend on protocol-sensitive metrics or qualitative evidence.
- The EPG extension clearly improves foreground localization metrics, but the evidence is not yet enough to conclude that it improves saliency faithfulness or prompt-specific interpretability.

**Requested Changes:**

- The paper often reports only directional reproducibility rather than numerical reproducibility. Statements such as “fully validating” Claim 1 are too strong given that absolute localization metrics, Network Dissection counts, and ROAR curves differ substantially from the original paper.
- Claim 1 is only partially reproduced. The paper shows that retrained AFT4 and the author-released AFT4 checkpoint match each other under the authors' new evaluation pipeline, which supports model-level reproducibility. However, the absolute values differ from the original paper, and the explanation is attributed to thresholding, upsampling, and hit-radius choices. These choices can affect baseline and AFT models differently, so the paper should not assume that the relative improvements are unaffected without additional sensitivity analysis.
- The Grad-CAM part of Claim 1 is weak. The paper finds Grad-CAM unreliable for CLIP-style encoders and treats it as inconclusive. That is a reasonable observation, but it means the original claim about both Simple Gradient and Grad-CAM is not fully reproduced. This should be stated more directly.
- Claim 2 is reproduced only in trend. The Network Dissection detector counts differ greatly from the original paper. This suggests that concept-alignment results are highly protocol-sensitive. The paper should provide stronger analysis of why these counts differ and whether the same conclusion holds under alternative concept vocabularies, thresholds, layers, and token choices.
- Claim 3 is supported mainly by qualitative evidence. The OOD and downstream transfer sections rely on visual comparisons rather than quantitative localization, deletion/insertion, or task-based metrics. This is too weak to strongly support the claim that interpretability improvements transfer across domains and downstream tasks.
- The zero-shot accuracy evaluation is under-specified and potentially unstable. It uses 1,000 random samples per dataset, but the paper does not report random seeds, confidence intervals, prompt templates, class-name choices, or sensitivity to sample selection.
- The adversarial evaluation needs more detail. The paper mentions APGD-CE and APGD-DLR attacks, but it does not fully specify the loss used for CLIP zero-shot classification, whether the text prompts are fixed, how targeted attacks are constructed, or whether evaluation follows a standard AutoAttack configuration.
- The EPG extension is promising but not yet well separated from the evaluation metric. The method directly trains saliency mass to lie inside foreground masks and is then evaluated mostly with foreground-mask localization metrics. This makes the improvement partly expected. Stronger evidence would require metrics less directly aligned with the training objective, such as deletion/insertion, ROAR after EPG fine-tuning, counterfactual background changes, or prompt-specific object localization.
- The theoretical discussion of EPG is interesting but somewhat disconnected from the empirical claims. The appendix gives a Hessian and background-adversary interpretation, but the main empirical evidence does not test the implied invariances directly. The theory should either be shortened or tied to a concrete experiment.

---

### Review · Reviewer_h5uz · 2026-07-21

**Summary Of Contributions:**

This paper attempts to reproduce a previous study by Gong et al. (2025), which proposed adversarial fine-tuning (AFT) to improve the visual interpretability of CLIP. The authors independently verify the four central claims of the original paper, including saliency map quality, neuron-concept alignment, transferability, and the accuracy/robustness trade-off. Additionally, they implement several evaluation protocols missing from the original repository, including localization metrics, concept-alignment evaluation, adversarial robustness, retrieval robustness, and ROAR. Beyond reproduction, the paper proposes a saliency-guided regularization extension based on an Energy Pointing Game loss, which uses object masks to explicitly encourage gradient saliency maps to align with foreground regions.

**Strengths:**
- The paper studies the reproducibility of a recent CLIP interpretability method.
- Missing evaluation protocols are carefully detailed to demonstrate a useful reimplementation effort.
- The paper is transparent that some claims are reproduced directionally rather than numerically exactly.
- The proposed saliency-guided extension is straightforward and shows improved concept localization.

**Weaknesses:**
- The first contribution, reproducing the main claims of Gong et al. (2025), mainly confirms trends already reported in a recent ICLR paper. This does not reveal a major failure of the original method, or identify new scientific insight that changes our understanding of adversarial fine-tuning or interpretability.
- The second contribution, implementing missing evaluation protocols, is valuable as an engineering effort but marginal as a standalone research contribution. Missing thresholds, upsampling rules, dataset subsets, and Pointing Game details could potentially be clarified by contacting the original authors or requesting the missing scripts.
- Because reproduction is the main contribution, the paper should summarize each claim explicitly as fully reproduced, directionally reproduced, partially reproduced, or not reproduced.
- The proposed saliency-guided regularization extension is incremental. Improvements on mask-based localization metrics are partly expected from the supervised objective itself. Additionally, the proposed extension requires annotated foreground masks and second-order optimization through input gradients. This limits applicability in domains where object masks or equivalent concept annotations are unavailable or expensive.
- The extension is validated only in a narrow COCO-based fine-tuning setting. The paper does not establish whether the learned saliency behavior transfers to another dataset or domain. Evaluation on ImageNet-S after COCO fine-tuning is helpful, but it does not fully answer whether the method works when directly trained with another dataset’s object/concept annotations or whether its generalizes to substantially different domains.

**Audience:**

No

**Audience Explanation:**

The paper may be useful to a narrow group of researchers working directly on Gong et al. (2025), adversarially fine-tuned CLIP, or saliency evaluation. However, I am not convinced that the findings provide sufficient new research insight for the TMLR audience. The first two contributions mainly reproduce an existing recent paper and reconstruct missing evaluation infrastructure. These efforts are useful for reproducibility, but they do not substantially advance a research question or introduce a broadly reusable methodological insight. The third contribution is a supervised mask-alignment loss whose outcome is intuitive: explicitly supervising saliency with object masks improves mask-based localization.

**Broader Impact Concerns:**

I do not see any broader-impact concerns.

**Claims And Evidence:**

Yes

**Claims Explanation:**

The paper generally supports the claims it makes within its stated scope. The authors reproduce the main directional trends of the original AFT work, and clearly acknowledge that some absolute values differ from the original paper. The benefits of saliency-guided extension is also supported, where increasing the EPG-loss weight improves Simple Gradient localization metrics and partially transfers to Grad-ECLIP, while robustness and clean accuracy decrease moderately.

However, my concern is that the contributions are incremental and narrow: two contributions are mainly reproduction/reimplementation, while the new supervised extension is evaluated in a limited setting and directly optimizes metrics closely related to its objective.

**Requested Changes:**

- **Add a claim-by-claim reproducibility summary:** The introduction or conclusion should contain a concise table covering all four original claims. For each claim, the authors should report whether it was fully reproduced, directionally reproduced, partially reproduced, or not reproduced, together with the relevant experiment and primary source of discrepancy.
- **Clarify the scientific insight from the reproduction:** The authors should explain what is learned beyond confirming previously reported trends and rebuilding missing evaluation scripts. In particular, does the reproduction reveal any general lesson about interpretability evaluation, robustness, or sensitivity to protocol choices that extends beyond the original AFT paper?
- **Evaluate the supervised extension on an additional dataset:** The authors should conduct at least one of the following: (i) directly fine-tune the extension on another dataset with corresponding object, concept, or segmentation labels and evaluate it there; (ii) evaluate generalization to a substantially different annotated dataset without additional saliency supervision. This experiment would clarify whether the extension learns general saliency alignment or mainly adapts to COCO-style object masks.
- **Clarify the supervision and applicability trade-off:** The paper should explicitly contrast unsupervised AFT with the proposed supervised extension and discuss the requirement for dense masks, the possibility of weaker concept-level supervision, and the computational cost of second-order optimization.